# Quantum-inspired tempering for ground state approximation using artificial neural networks

Tameem Albash[1,2,3*], Conor Smith[1,3], Quinn Campbell[4] and Andrew D. Baczewski[2,3,4]

**1** Department of Electrical and Computer Engineering,
University of New Mexico, Albuquerque, NM 87131, USA
**2** Department of Physics and Astronomy,
University of New Mexico,Albuquerque, NM 87131, USA
**3** Center for Quantum Information and Control (CQuIC),
University of New Mexico, Albuquerque, NM 87131, USA
**4** Quantum Algorithms and Applications Collaboratory,
Sandia National Laboratories, Albuquerque NM 87185, USA

⋆ talbash@unm.edu

## Abstract

A large body of work has demonstrated that parameterized artificial neural networks (ANNs) can efficiently describe ground states of numerous interesting quantum many-body Hamiltonians. However, the standard variational algorithms used to update or train the ANN parameters can get trapped in local minima, especially for frustrated systems and even if the representation is sufficiently expressive. We propose a parallel tempering method that facilitates escape from such local minima. This methods involves training multiple ANNs independently, with each simulation governed by a Hamiltonian with a different "driver" strength, in analogy to quantum parallel tempering, and it incorporates an update step into the training that allows for the exchange of neighboring ANN configurations. We study instances from two classes of Hamiltonians to demonstrate the utility of our approach using Restricted Boltzmann Machines as our parameterized ANN. The first instance is based on a permutation-invariant Hamiltonian whose landscape stymies the standard training algorithm by drawing it increasingly to a false local minimum. The second instance is four hydrogen atoms arranged in a rectangle, which is an instance of the second quantized electronic structure Hamiltonian discretized using Gaussian basis functions. We study this problem in a minimal basis set, which exhibits false minima that can trap the standard variational algorithm despite the problem's small size. We show that augmenting the training with quantum parallel tempering becomes useful to finding good approximations to the ground states of these problem instances.



# 1  Introduction

Approximating the ground state of a many-body quantum Hamiltonian is a computational task at the heart of many problems in the physical sciences. Tackling this problem requires both an efficient representation of the wave function and an algorithm for optimizing the wave function to minimize its energy whose individual update steps are efficient. A variety of methods have been developed, each with its advantages and limitations. Quantum Monte Carlo (QMC) [1, 2] methods are generically useful in the absence of the sign problem [3], although there are QMC approaches that try to mitigate the bottlenecks associated with the sign problem [4–6]. Methods based on Matrix Product States (MPS) [7–12] have been extremely successful at addressing systems in one dimension and methods based on Projected Entangled Pair States (PEPS) [13,14] have allowed the study of certain systems in two dimensions. More recently, artificial neural networks (ANNs) have been used as ansätze for quantum ground states [15], with the advantages of being agnostic to the sign problem and able to take as input Hamiltonians with arbitrary connectivity. While there have been important demonstrations of the utility of this approach [16–18], it still suffers when studying frustrated systems by getting stuck in the many local minima of a rugged optimization landscape [19].

Parallel tempering (PT) [20–22] is a Monte Carlo method that has been extensively used to help explore the rugged energy landscapes of spin glasses [23]. In the typical PT approach, multiple "replicas" of the system evolve independently at different temperatures and hence in different free energy landscapes. The exchange of configurations between higher and lower temperature replicas facilitates escape from local minima. There is nothing special about using

temperature to alter the landscape, and other physically motivated choices can be made. In quantum parallel tempering (QPT) [24,25], replicas are evolved according to different Hamiltonians, where the strength of a driver Hamiltonian, e.g. a uniform transverse field, is used to control the landscape the replicas are evolving in instead of the temperature.

In this paper we combine the methods of QPT and the training of ANNs to approximate ground states of quantum spin systems. For simplicity of presentation, we focus on a specific choice of ANN, the restricted Boltzmann machine (RBM), and we demonstrate how a QPT method implemented in addition to the standard training can help overcome barriers in energy landscape. We do this using two examples. The first is based on a permutation-invariant Hamiltonian that is designed to trap the standard training approach in a false minimum. In this case, the QPT provides a clear advantage over simply repeating the standard algorithm starting from independent initial points. The second is based on a molecular system of four hydrogen atoms arranged in a rectangle with an angle $\theta$ between adjacent hydrogen atoms that we can vary. The fermionic second quantized Hamiltonian is mapped to a system of qubits using the Jordan-Wigner transformation [26], and we apply the ANN training methods to the qubit Hamiltonian [27]. We find that the hardness of approximating the ground state of this system depends on different aspects, such as the choice of angles and whether we constrain the training to a fixed magnetization sector corresponding to a fixed particle number, and we find that QPT can provide a computational advantage in the cases where the standard approach struggles to find a good approximation of the ground state.

The paper is organized as follows. In Section 2, we review the standard training approach of the RBM ansatz for quantum wave functions in addition to how we introduce QPT to it. In Section 3, we introduce our two problem classes and compare the performance of the standard training approach to our QPT augmented approach. In Section 4, we conclude with open questions and future directions of our work.

## 2 Overview of the Methodology

We consider an unnormalized parameterized wave function expressed in the computational basis $|\psi(\alpha)\rangle = \sum_{x \in \{0,1\}^n} \psi_x(\alpha)|x\rangle$, where $|x\rangle \equiv |x_n\rangle \otimes \cdots \otimes |x_1\rangle$ denotes the $n$-qubit computational basis state with $\sigma_i^z|x_i\rangle = (1 - 2x_i)|x_i\rangle = v_i|x_i\rangle$ and $\alpha$ denotes the set of $K$ complex variational parameters. Here we restrict our attention to the case where $\psi_x(\alpha)$ is the weight associated with a RBM with parameters $\alpha = \{a_i, b_\mu, W_{i\mu}\}_{i=1,\mu=1}^{n,m}$ and with $x$ as the value of its input layer,

$$\psi_x(\alpha) = e^{\sum_{i=1}^n a_i v_i} \prod_{\mu=1}^m \cosh\left(b_\mu + \sum_{i=1}^n W_{i\mu} v_i\right), \tag{1}$$

where $v_i \in \{-1, 1\}$. As long as $m$ scales polynomially with $n$, the RBM representation provides an efficient representation of the quantum state $|\psi(\alpha)\rangle$.

Given a many-body Hamiltonian $H$ defined on $n$ qubits, our aim is to find an approximate RBM representation to the ground state of $H$. Therefore, our optimization problem is to identify a set of parameters $\alpha_*$ such that

$$\alpha_* \in \underset{\alpha}{\mathrm{argmin}} \; \frac{\langle \psi(\alpha)|H|\psi(\alpha)\rangle}{\langle \psi(\alpha)|\psi(\alpha)\rangle}. \tag{2}$$

The cost function on the right hand side defines a multi-dimensional parameter landscape, and if the number of variational parameters is sufficiently large, the global minimum of this landscape gives an excellent approximation of the ground state.

This optimization problem is solved using Stochastic Reconfiguration (SR) [28,29], which updates the parameters as

$$\alpha'_k = \alpha_k + \delta\alpha_k = \alpha_k - \gamma \sum_{k'} (S(\alpha))^{-1}_{kk'} F_{k'}(\alpha), \tag{3}$$

where $S_{kk'}(\alpha)$ are the matrix elements of the covariance matrix and $F_k(\alpha)$ are the elements of the force vector defined as,

$$S_{kk'}(\alpha) = \langle O_k(\alpha)^\dagger O_{k'}(\alpha)\rangle - \langle O_k(\alpha)^\dagger\rangle\langle O_{k'}(\alpha)\rangle, \tag{4a}$$

$$F_k(\alpha) = \langle O_k(\alpha)^\dagger H\rangle - \langle O_k(\alpha)^\dagger\rangle\langle H\rangle, \tag{4b}$$

where the expectation values are taken with respect to the state $|\psi(\alpha)\rangle$ and where $O_k(\alpha)$ is the operator associated with the gradient of the state $|\psi(\alpha)\rangle$ with respect to the variational parameter $\alpha_k$,

$$O_k(\alpha) = \sum_{x=\{0,1\}^n} \frac{1}{\psi_x(\alpha)} \frac{\partial}{\partial\alpha_k}\psi_x(\alpha)|x\rangle\langle x|. \tag{5}$$

Since $S$ may not be invertible, $S^{-1}$ is strictly speaking the Moore-Penrose pseudoinverse [30, 31]. We review the derivations of these formulas in Appendix A.

The parameter $\gamma$ sets the learning rate, and we choose it adaptively using a second-order Runge-Kutta (RK) method [17, 19, 32]. We review this adaptive procedure in Appendix B. While the RK method incurs a computational overhead because it requires multiple estimates of the covariance matrix and force vector, the combination of RK with SR generally results in more accurate results with fewer updates in total compared to other optimization schemes [19] such as ADAM [33] with SR or stochastic gradient descent, and it eliminates the need to specify a learning rate. In this regard, the RK with SR optimization approach to train the RBM sets a competitive baseline for us to improve upon.

Calculating the covariance matrix $S$ and force vector $F$ requires expectation values of the observables associated with the set $\{O_k\}$ and $H$. These can be estimated using Monte Carlo importance sampling [15], but in this work we will restrict ourselves to evaluating these expectation values exactly. The reason for using exact sampling is to eliminate the role of finite sampling in the training. We give details of how to calculate the necessary expectation values in Appendix C.

With $S$ and $F$ in hand, the final step to implement Eq. (3) is to calculate the pseudoinverse of $S$. In what follows we calculate the inverse using an explicit regularization where we replace the diagonal elements of $S$ by [15]

$$S_{kk} \to S_{kk}^{\text{reg}} = S_{kk}(1 + \lambda(p)), \tag{6}$$

where $p$ denotes the update step and $\lambda(p) = \max(\lambda_0 b^p, \lambda_{\min})$ with $\lambda_0 = 100, b = 0.9, \lambda_{\min} = 10^{-4}$. We use the MINRES-QLP [34] with a maximum of $10^3$ iterations and a stopping tolerance of $10^{-6}$ to solve the linear system of equations. The choice of these parameters can affect the performance of SR to find a description of the ground state, and we have not tried to optimize these parameters for our problem; our aim is to compare the *relative* performance of the overall algorithm with and without QPT using the same set of parameters otherwise.

The SR algorithm can be shown [35] to be a generalization of the Riemannian natural gradient [36], so like other gradient descent algorithms it is susceptible to being trapped in local minima of the energy landscape. Finite sampling and the resulting fluctuations in $S$ and $F$ can be a source of noise that can help the training escape local minima [37], but it can also hinder an accurate determination of the gradient and hence prevent the training from finding

high-quality solutions. Motivated by the success of PT [20–22] in the study of equilibration in spin systems, we propose the following optimization algorithm that is inspired by QPT [24,25]. We perform the usual training on $N$ RBMs, each called a replica in the parlance of PT. Each training is done with a different training Hamiltonian, where the Hamiltonian for the $r$th replica is given as

$$H(\Gamma_r) = H_{\mathrm{T}} + \Gamma_r H_{\mathrm{M}}, \tag{7}$$

where $H_{\mathrm{T}}$ is the target Hamiltonian whose ground state we are actually after and $H_{\mathrm{M}}$ is a suitably chosen mixer Hamiltonian with a trivial ground state, preferably the one with a uniform superposition over the relevant basis states. This choice of training Hamiltonian is not unique, and we give an alternative formulation in Appendix D. For example, in the absence of any constraints, the mixer can be taken to be the uniform transverse field Hamiltonian, $H_{\mathrm{M}} = H_0 \sum_{i=1}^{n} \frac{1}{2} \left( \mathbb{1} - \sigma_i^x \right)$, where $H_0$ is a characteristic energy scale of $H_{\mathrm{T}}$. In this case, for replicas with $\Gamma_r \gg 1$ the training Hamiltonian is dominated by the transverse field, and the ground state is the uniform superposition state. This is analogous to the high-temperature limit in "standard" PT, where the different replicas operate at different temperatures but with the same Hamiltonian. For replicas with $\Gamma_r$ close to 0, the training Hamiltonian is dominated by the target Hamiltonian, and the ground state is close to the target ground state. This is analogous to the low-temperature limit in standard parallel tempering. The parameter $\Gamma_r$ thus allows us to interpolate between the target parameter landscape at $\Gamma_r = 0$ and a much simpler parameter landscape at $\Gamma_r \gg 1$. We can therefore think of $\Gamma_r$ as controlling the effective temperature of the simulation. We note that using Eq. (7) as the training Hamiltonian and using $\Gamma_r$ as the temperature is similar to schemes that use the free energy to train the ANN [18,38,39], where instead of using the entropy we use $\langle H_{\mathrm{T}} \rangle$. We leave it to future work to directly compare these two approaches.

We choose our distribution of $\Gamma_r$ to be a cubic function of the replica indices,

$$\Gamma_r = 10 \left( \frac{r}{N-1} \right)^3, \quad r = 0, \dots, N-1. \tag{8}$$

For our simulations, a $\Gamma_{N-1} = 10$ value was sufficient for the $H_{\mathrm{M}}$ to dominate over $H_{\mathrm{T}}$. We emphasize that the choice of the distribution of $\Gamma_r$ values can strongly influence the performance of the QPT algorithm; for example, choosing a linear function for our simulations resulted in very poor performance.

Our algorithm proceeds as follows. The $N$ replicas (indexed by $r = 0, \dots, N-1$) are initialized randomly with both the real and imaginary components of the parameters $\alpha_r$ chosen from a Gaussian distribution with mean 0 and standard deviation $10^{-2}$. We perform 10 updates of SR for all replicas, configuration swap between replicas $r$ and $r+1$ for $r$ even, 10 updates of SR for all replicas, and finally a configuration swap between replicas $r$ and $r+1$ for $r$ odd. The choice of the number of updates of SR between configuration swaps was not optimized, and other choices may improve performance. This process is repeated until a total of $u$ updates are performed. The deterministic even-odd pattern for configuration swaps is a typical update pattern in PT that helps improve the diffusion of replicas [40,41]. For every update, we calculate the expectation value of the target Hamiltonian for each replica, which we denote by

$$E_r \equiv \langle \psi(\alpha_r) | H_{\mathrm{T}} | \psi(\alpha_r) \rangle. \tag{9}$$

We denote the running average of an expectation value of an observable $O$ over 10 updates of SR by $\overline{\langle O \rangle}$, which we will use to calculate the probabilities for configuration swaps.

The final issue we must address about the QPT algorithm is how to perform the swap updates between the $N$ replicas. We begin by reminding ourselves how the swaps are performed in the case where each of the $N$ replicas is evolving according to a Markov Chain Monte Carlo simulation that is ergodic and has a unique equilibrium distribution. Assuming

a finite state space, the $i$-th configuration of replica $r$ has an equilibrium probability given by $\Pi(\Gamma_r) = W_i(\Gamma_r)/\sum_j W_j(\Gamma_r)$, where $W_i(\Gamma_r)$ is the unnormalized weight of the $i$-th configuration. The swap updates between replicas then occurs with a probability $p$ chosen to satisfy the detailed balance condition; specifically, if replica $r$ is in configuration $i$ and replica $r+1$ is in configuration $i'$, then the probability of a swap between the two is taken to be:

$$p_{r \leftrightarrow r+1} = \min\left(1, \frac{W_i(\Gamma_{r+1})W_{i'}(\Gamma_r)}{W_i(\Gamma_r)W_{i'}(\Gamma_{r+1})}\right). \tag{10}$$

While we may think of the SR algorithm as implementing imaginary-time evolution in a fixed subspace (see Appendix A), we are not aware of a way to implement this evolution in terms of an ergodic MCMC simulation that eventually converges to an equilibrium distribution. Therefore we are not able to use the form in Eq. (10), and instead we will use physical intuition to pick a swap update probability. This choice is not unique, and we expect different choices to perform differently.

With these concerns in mind, we propose the following update. For $\Gamma_r = 0 (r = 0)$, we expect the ANN to reach a good approximation of an eigenstate of $H_{\mathrm{T}}$, assuming the number ANN parameters is sufficiently large and the ANN is sufficiently expressive. Similarly, for replicas with $\Gamma_r \gg 1$ we expect the ANN to be a good approximation of an eigenstate of $H_{\mathrm{M}}$. Therefore, as we have already mentioned, we can think of $\Gamma_r$ as being a measure of a "noise" introduced, and we can think of $\Gamma_r$ as being proportional to the temperature of the replica. We thus propose the following probability rule for swap updates between neighboring pairs $r$ and $r+1$:

$$p_{r \leftrightarrow r+1} = \min\left(1, \exp\left[\left(\overline{E_r} - \overline{E_{r+1}}\right)\left(\frac{1}{H_0\Gamma_r} - \frac{1}{H_0\Gamma_{r+1}}\right)\right]\right). \tag{11}$$

This choice for the probability rule is a heuristic choice motivated by the Metropolis update [42] for the standard PT algorithm [20–22], but in no way do we claim that it satisfies detailed balance [43]. The use of the target energy comes from the desire to move configurations with lower target energies to smaller $r$. For our temperature, we use $\Gamma_r$ multiplied by some relevant energy scale $H_0$ as motivated by our discussion above. We use $\overline{E_r}$ as opposed to simply $E_r$ in order to average out any fluctuations in the estimation of the target energy during the ANN training.

In what follows, we define success as finding a configuration below some relative error $\epsilon$ of the energy given by

$$\epsilon = \left|\frac{\langle E \rangle - E_{\mathrm{GS}}}{E_{\mathrm{GS}}}\right|, \tag{12}$$

where $E_{\mathrm{GS}}$ is the true ground state energy. To determine whether QPT can provide practical advantages over its standard counterpart, we use the time-to-epsilon (TT$\epsilon$) metric [44, 45], which is the time required to find a solution below a target relative error $\epsilon$ at least once with probability 0.99. The TT$\epsilon$ is given by

$$\mathrm{TT}\epsilon = \frac{N}{N_{\mathrm{max}}}u\frac{\ln(1-0.99)}{\ln(1-p_{\mathrm{S}}(u))}, \tag{13}$$

where $p_{\mathrm{S}}(u)$ is the probability of reaching the target $\epsilon$ within $u$ steps of the algorithms, such that the factor $\frac{\ln(1-0.99)}{\ln(1-p_{\mathrm{S}}(u))}$ counts the number of repetitions of the algorithm needed to ensure the target $\epsilon$ is reached with probability 0.99 at least once. $N_{\mathrm{max}}$ is the largest number of QPT replicas we will use in our simulations, so that the factor $N/N_{\mathrm{max}}$ takes into account the reduction in computational cost possible by parallelization by exchanging replicas $N$ with independent runs of the algorithm. For every pair of values $(n, N)$, we identify the value of $u$ that minimizes the TT$\epsilon$, and we choose this as the cost of running the algorithm. Using this

approach we can determine unambiguously when it becomes more advantageous to use QPT versus independent runs of the algorithm. We emphasize that this metric does not account for the extra overhead associated with the communication needed between replicas to implement the swap update, but we expect this to be minimal compared to the cost of finding a high quality solution for hard problems.

## 3 Results

### 3.1 Precipice Problem

In order to illustrate the computational viability of the QPT approach, we study a problem class of qubit Hamiltonians that are invariant under any permutation of the qubits. Specifically, we take our target Hamiltonian to be of the form

$$H_{\mathrm{T}} = \frac{1}{2}(1-s)\sum_{i=1}^{n}\left(\mathbb{1} - \sigma_i^x\right) + s\sum_{x\in\{0,1\}^n} f(x)|x\rangle\langle x|, \tag{14}$$

where $n$ is the total number of qubits and $f(x)$ is a function that only depends on the Hamming weight of the bit-string $x$. We can therefore label the energy eigenstates of $H_{\mathrm{T}}$ in terms of their symmetry properties under qubit permutation. We take the mixer Hamiltonian to be given by the uniform transverse field Hamiltonian,

$$H_{\mathrm{M}} = \frac{H_0}{2}\sum_{i=1}^{n}\left(\mathbb{1} - \sigma_i^x\right), \tag{15}$$

which is also qubit permutation invariant and we set $H_0 = 1$. Because the Hamiltonian is stoquastic [46, 47], the ground state of this Hamiltonian can be expressed entirely with non-negative amplitudes, so it must be in the completely symmetric subspace, which has dimension $n+1$. We can enforce this symmetry in our RBM ansatz

$$\psi_x(\alpha) = e^{a\sum_{i=1}^{n} v_i}\prod_{\mu=1}^{m}\cosh\left(b_\mu + W_\mu\sum_{i=1}^{n} v_i\right), \tag{16}$$

where $v_i \equiv (1-2x_i) \in \{-1, 1\}$. This ansatz only depends on the total magnetization of the spin configuration $v$, or equivalently on the Hamming weight of the bit-string $x$. This will enforce that all equal Hamming weight bit-strings will have the same amplitude. This representation has only $K = 1 + 2m$ parameters. In principle these parameters need only be purely real to capture the ground state, but we will allow the training to treat them as complex parameters.

In what follows, we will focus on a function $f(x)$ first introduced in Ref. [48]

$$f(x) = \begin{cases} -1, & \text{if } x = 1\cdots1, \\ w(x), & \text{otherwise}, \end{cases} \tag{17}$$

where $w(x)$ denotes the Hamming weight of the bit-string $x$. We call this problem the "Precipice" problem because of the form of $f(x)$, since it increases linearly with Hamming weight until reaching Hamming weight $n$ where it plummets to a value of $-1$.

For concreteness, we pick $s = 4/5$ for our target Hamiltonian in Eq. (14). For this value of $s$, $H_T$ is dominated by its diagonal component, corresponding to the second term in Eq. (14). The ground state of the corresponding target Hamiltonian $H_{\mathrm{T}}$ has most but not all of its weight on the all-one bit-string, which is the bit-string that minimizes $f(x)$. On the other hand, the first excited state of $H_{\mathrm{T}}$ has most of its weight on the false minimum of $f(x)$ given by the

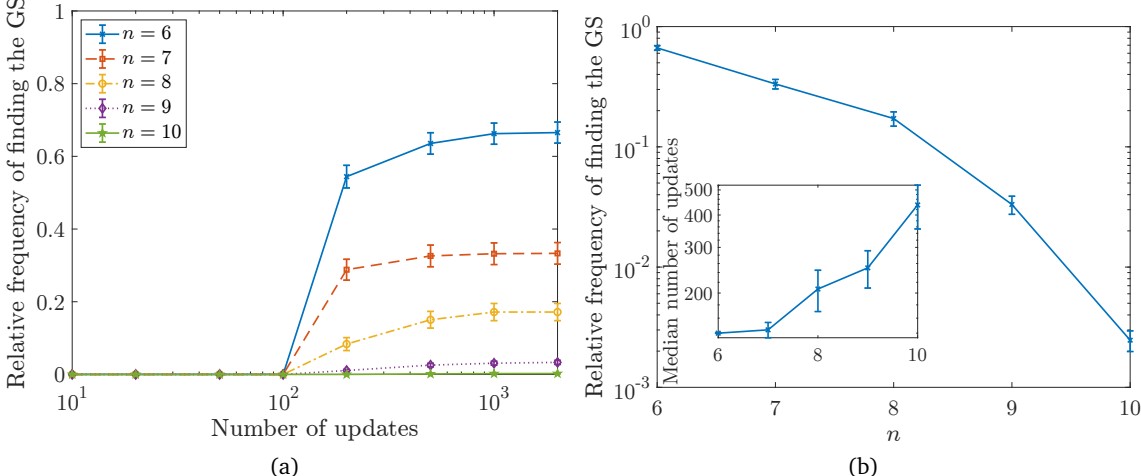

Figure 1: Relative frequency of finding the ground state for the Precipice problem to within a relative error of $\epsilon = 0.1$ for a random initialization of a trial RBM ansatz without using QPT as a function of (a) number of updates and (b) system size $n$. We used $10^3$ independent trials for $n = 6, 7, 8$, $4 \times 10^3$ for $n = 9$ and $4 \times 10^4$ for $n = 10$. The error bars correspond to the 95% confidence interval calculated using a bootstrap with $10^3$ resamplings of the data. We describe this procedure in Appendix E. Inset: Median number of updates to reach a relative error of $\epsilon = 0.1$ when the algorithm succeeds.

all-zero bit-string. The Hamiltonian $H_T$ has the feature that annealing from $\Gamma \gg 1$ to $\Gamma = 0$ (Eq. (7)) encounters an exponentially closing ground state energy gap as a function of the system size. One can understand this intuitively as arising from the system requiring to tunnel between the two configurations above (the state with most of its weight on the all-one bit string and the state with most of its weight on the all-zero bit string) that requires flipping $n$ spins [49].

This feature of the energy spectrum stymies the SR algorithm. Whether it will find the the correct global minimum depends entirely on where the RBM parameters are initialized, because once it reaches the false minimum associated with the first excited state, the SR updates do not provide a means to escape this local minimum and reach the global minimum. This is where our QPT implementation helps, as it provides a mechanism for RBM configurations with weight on the all-one bit-string to be reached and guide the training to the true ground state. Since our aim is to distinguish between the two minima that the algorithm can get stuck in, the choice of $\epsilon$ is not crucial as long as it is sufficiently small. We therefore fix it to $\epsilon = 10^{-1}$, which is enough to distinguish between the two lowest energy eigenstates.

We begin by analyzing the behavior of the standard algorithm without QPT. We show in Fig. 1 how the probability of finding the ground state within a relative error of $\epsilon$ behaves as a function of system size. We see that the probability of finding the ground state saturates to a value less than 1 as a function of the number of SR updates, and the saturation value decreases faster than exponentially with system size. This behavior indicates that once the system reaches the false minimum it is trapped there without the possibility of escape, and the probability of initializing at a point where the SR updates guide you to the correct minimum decays faster than exponentially.

To show how QPT boosts the probability of finding the ground state, we show in Fig. 2 how with increasing number of updates in QPT we can increase the probability of finding the ground state. In order to compare these results to the standard approach without QPT,

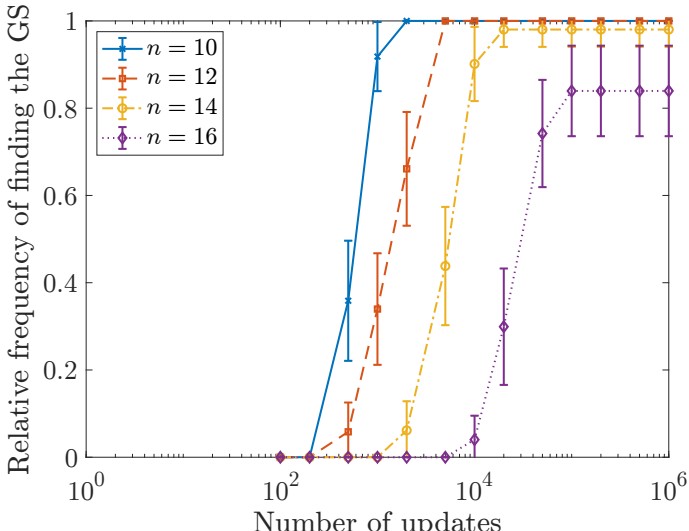

Figure 2: Relative frequency of finding the ground state for the Precipice problem to within $\epsilon = 10^{-1}$ with increasing number of updates in QPT for 50 independent simulations. Here we use $N = 10$ replicas for all QPT simulations.

we use the results at $n = 10$ as a benchmark. In the standard approach, the probability of finding the ground state is approximately $2.5 \times 10^{-3}$ with a median number of steps of 440 (Fig. 1). Therefore, to guarantee that we find the ground state at least once with probability 0.99, we would need to perform 1840 independent trials. With QPT, we can achieve almost a probability of 1 of finding the ground state using less than 500 steps and 10 replicas with a single simulation. Therefore, we already see a significant advantage for using QPT in this model.

To illustrate how the QPT algorithm is helping, we show in Fig. 3 an example of a random walk for one of our simulation trials. In Fig. 3a, we see that both replicas at $r = 0$ and $r = 1$ very quickly converge to the first excited state and only after approximately 1500 network updates does the $r = 0$ replica find the ground state. As we can see in Fig. 3b, the transition to the ground state is precipitated by a swap between the $r = 0$ and $r = 1$ configurations. While the configuration at $r = 0$ remains unchanged for a large fraction of the earlier updates, the configuration at $r = 1$ is swapped multiple times with other configurations, allowing it to explore the landscape of possible solutions.

Finally, we show our results for the TT$\epsilon$ in Fig. 4 with $N_{\max} = 40$. We see that at small system sizes ($< 8$), using the standard approach ($N = 1$) is more advantageous than incurring the added computational cost of QPT. Here the probability of reaching the ground state is sufficiently high (even if it is not close to 1 as seen in Fig. 1) that running multiple independent runs of the algorithm is the optimal strategy. However beyond system sizes of $> 8$, we see a significant advantage to using QPT. In this case, the optimal strategy is to use QPT with more updates to achieve higher probabilities of reaching the ground state.

We note that for intermediate values of $n$ we see similar performance for $N = 10$ and $N = 20$, indicating that increasing the number of replicas does not improve the success probability enough to outweigh the cost of simulating additional replicas. However, we also see that at larger system sizes it does become more advantageous to use more replicas. For example, at $n = 17$ our simulations with $N = 10$ replicas fail to find the ground state, and at $n = 18$ our simulations with $N = 20$ replicas fail to find the ground state. Because increasing $N$ changes the distribution of $\Gamma_r$ values and includes more replicas at smaller $\Gamma_r$ values, the need to increase $N$ to find the ground state at larger system sizes indicates the importance of optimizing

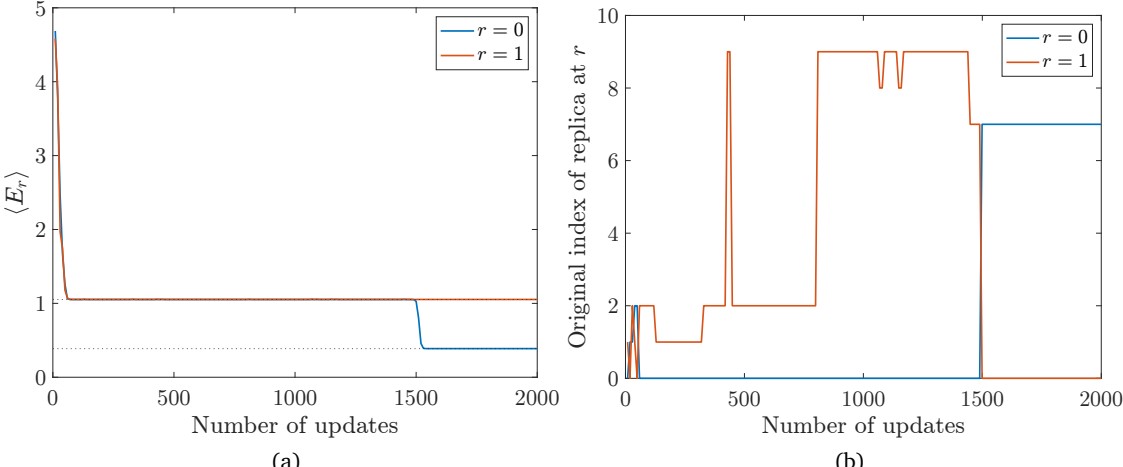

Figure 3: An example of a random walk for a single simulation of the Precipice problem with $n = 12$ and $N = 10$. In (a), we show the target energy (Eq. (9)) of the configuration at $r = 0$ and $r = 1$ during the simulation. The dotted lines correspond to the ground state energy and first excited state energy for the target Hamiltonian. In (b), we label each configuration by its original replica index at initialization, and we show which index is at position $r = 0$ and $r = 1$ during the simulation. This allows us to monitor how configurations walk on the space of $\Gamma_r$.

this distribution to get the best results.

## 3.2 H$_4$ Rectangle

The second system we study is that of four hydrogen atoms arranged in a rectangle with an angle $\theta$ between adjacent atoms. This is an example of a molecular system that is small enough to be exactly solved that includes a type of configurational degeneracy that gives some approximate quantum chemistry algorithms trouble [50, 51]. We consider a particular encoding of the second quantized Hamiltonian into 8 qubits; further details of this system and how we construct the qubit Hamiltonian are given in Appendix F.

The ground state of the 8-qubit Hamiltonian has non-zero amplitudes only on computational basis states with Hamming Weight four or equivalently with four fermions or zero $z$-magnetization. This observation motivates two different RBM ansätze: (1) we restrict the input of the RBM to Hamming-weight-four computational basis states and assume that all other inputs give zero amplitude, and (2) we allow all inputs to the RBM and the training of the RBM must find the Hamming weight-four subspace. While the first case is a more efficient representation for the H$_4$ Rectangle, we still consider both ansätze since they could be relevant in different contexts for fermionic simulation beyond the H$_4$ Rectangle: the first applies to cases where particle number is conserved, whereas the second applies to cases where the chemical potential is fixed.

In what follows, we fix our desired error to $\epsilon = 10^{-3}$. The reason for this smaller choice of relative error, compared to the Precipice example, is that this problem requires a relative error less than $6 \times 10^{-3}$ for $\theta = 90°$ to begin distinguishing the two lowest energy eigenstates. So we choose an $\epsilon$ that is sufficiently small to ensure that we are finding a high quality approximation of the ground state. For the simulations, we use a up to $10^4$ updates.

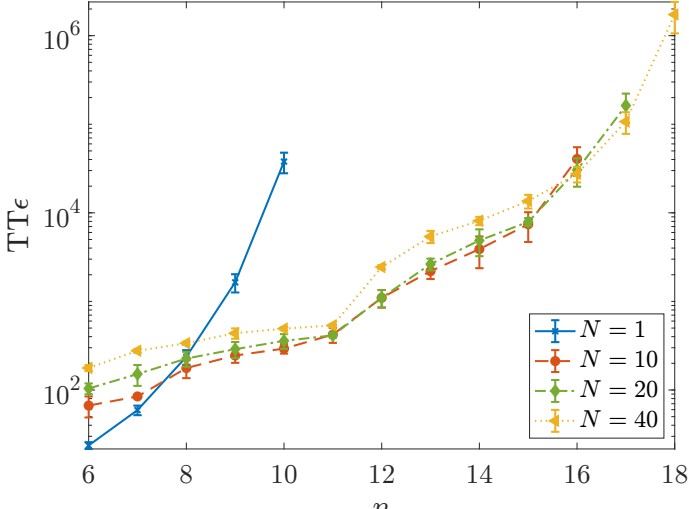

Figure 4: Optimum time to epsilon to reach a relative error of $\epsilon = 10^{-1}$ for the Precipice problem as a function of system size $n$ for the Precipice problem with different number of QPT replicas $N$ with $N_{\max} = 40$. The error bars correspond to the 95% confidence interval calculated using a bootstrap with $10^3$ resamplings of the data.

### 3.2.1 Fixed Hamming Weight

We first consider the case where the RBM is only sampled with Hamming-weight-four states. In this case, we choose a mixer Hamiltonian given by the ferromagnetic Heisenberg model with all-to-all connectivity,

$$H_{\mathrm{M}} = \frac{H_0}{2n} \sum_{i=1}^{n} \sum_{j=i+1}^{n} \left( \mathbb{1} - \sigma_i^x \sigma_j^x - \sigma_i^y \sigma_j^y - \sigma_i^z \sigma_j^z \right), \tag{18}$$

for which the (degenerate) ground state is the uniform superposition of fixed Hamming weight states with energy 0 and $H_0$ is the absolute magnitude of the largest coupling strength in the target Hamiltonian. This choice is made to ensure that the energy scales of $H_{\mathrm{M}}$ and $H_{\mathrm{T}}$ are comparable.

For the simulations, we find that using $m = 8$ hidden units is sufficient to represent the ground state with the required accuracy, so we use this value for all our training simulations with a fixed Hamming weight exact sampling. We show in Fig. 5 the results for the optimal TT$\epsilon$ for three different $\theta$ values, where we find rich behavior as a function of $\theta$. We find that for a sufficient number of replicas, the QPT approach demonstrates an advantage over the standard approach for angles $\theta = 80°$ and $85°$. For $\theta = 90°$, the ground state is found readily by the standard approach that the additional cost of running $N$ replicas does not introduce any benefit. The $\theta = 90°$ instance was found to be pathological for the coupled cluster doubles (CCD) approximation but not its variational counterpart in Ref. [51]. It is therefore unsurprising that a variational RBM ansatz that explores a similar Hilbert space captures the ground state well. Because the root of the CCD pathology is ultimately degeneracy of two of the dominant configurations, it seems like this degeneracy might provide an advantage for the RBM ansatz training that is worthy of further investigation.

We also note that this particular radius provides a uniquely easy energy landscape for the single-replica approach at 90°, and at different radii the 90° case is more difficult. In fact, among the instances considered, the instance that we are highlighting was found to be the one for which achieving an advantage with QPT was most difficult. At $\theta = 80°$ and $85°$, we

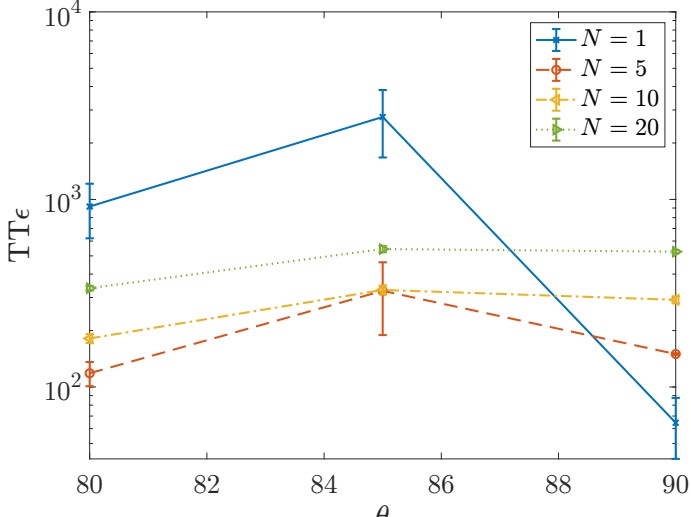

Figure 5: Optimum time to epsilon to reach a relative error of $\epsilon = 10^{-3}$ for the $H_4$ Rectangle at different angles with fixed Hamming weight exact sampling. We use $N_{\text{max}} = 20$. The error bars correspond to the 95% confidence interval calculated using a bootstrap with $10^3$ resamplings of the data. The lines are to guide the eye.

observe that $N = 5$, $N = 10$ and $N = 20$ replicas exhibit a definitive advantage over the single-replica approach.

### 3.2.2 No Fixed Hamming Weight

We now consider the case where the RBM is sampled with all possible computational basis states. In this case, we choose a mixer Hamiltonian to be the uniform transverse field:

$$H_{\text{M}} = \frac{H_0}{2} \sum_{i=1}^{n} \left( \mathbb{1} - \sigma_i^x \right), \tag{19}$$

where $H_0$ is again the absolute magnitude of the largest coupling strength in the target Hamiltonian.

For the simulations, we find that using $m = 48$ hidden units is sufficient to represent the ground state with the required accuracy but also allow the simulations with $N = 1$ to find the ground state with sufficient frequency, so we use this value for these simulations.[1] We show in Fig. 6 the results for the optimal TT$\epsilon$ for the same three values of $\theta \in \{80°, 85°, 90°\}$. While generally the problem of approximating the ground state is harder in this case compared to the fixed Hamming weight case, the advantage for QPT training with a sufficient number of replicas for all three angles is clear, indicating that the energy landscape is more efficiently searched using the QPT training in this case. This is a promising indication of the advantages of the QPT approach for more generic problems that may not exhibit any symmetries that can be used to restrict the wave function ansätze.

---

[1] We can actually use fewer hidden units (as low as $m = 16$) and still have an approximation of the ground state to the desired accuracy, but in this case we only find a good approximation to the ground state using our QPT method when using our simulation parameters.

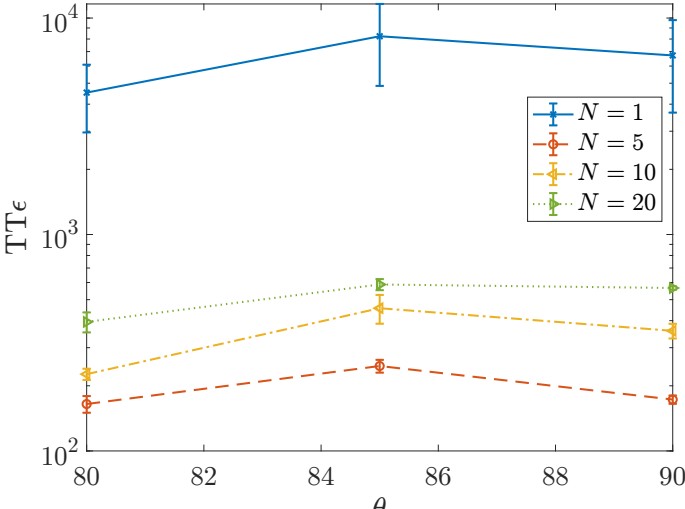

Figure 6: Optimum time to epsilon to reach a relative error of $\epsilon = 10^{-3}$ for the $H_4$ Rectangle at different angles with full exact sampling. We use $N_{max} = 20$. The error bars correspond to the 95% confidence interval calculated using a bootstrap with $10^3$ resamplings of the data. The lines are to guide the eye.

# 4 Conclusions

We have shown that using a QPT method as part of the training of ANNs to approximate ground states of quantum Hamiltonians can help overcome barriers in the energy landscape. To illustrate this, we simplify our analysis by using exact sampling in order to eliminate the role of fluctuations introduced by finite sampling. We demonstrated the utility of the QPT approach using two Hamiltonians with very different profiles. For the permutation invariant problem, the QPT approach demonstrated a clear advantage over repetitions of the standard algorithm even for small system sizes. A possible criticism of this problem is that it is described by a stoquastic Hamiltonian [46, 47]. Stoquastic Hamiltonians have ground state that can be described in terms of non-negative real amplitudes, and they admit efficient quantum-to-classical mappings [52]. While these properties do not necessarily mean there is an efficient algorithm to find the ground state, the existence of an efficient classical representation is often taken to mean that the ground states of stoquastic Hamiltonians are not as difficult to find as those of non-stoquastic Hamiltonians. We therefore consider a second problem based on four hydrogen atoms arranged in a rectangle with a variable angle between adjacent atoms, and we observe a clear advantage for the QPT approach at some angles where the standard approach struggled to find the ground state. These promising initial results indicate that using the QPT approach to train ANNs may be useful for a broad range of Hamiltonians with difficult to explore parameter landscapes.

We focused on using an RBM as our ANN ansatz, but the implementation of QPT is independent of the choice of ANN architecture and can be applied equally well to convolutional neural networks (CNNs) [16], to two disconnected real ANNs to represent the magnitude and phase of wave function amplitudes [19], or to group convolutional neural networks (GCNNs) [18], which have also been shown to be efficient at representing quantum states. This makes our proposed approach quite versatile, with only the added complexity of training multiple replicas of the ANNs and implementing configuration swaps between them.

We emphasize that we made several choices in this work for simplicity that might have been suboptimal for the performance of the QPT approach. For example, we chose a cubic distribution of the parameter $\Gamma$ for our replicas (Eq. (8)), and a suboptimal distribution of $\Gamma$

values around a bottleneck for the simulation does hinder the random walk of replicas [53]. We conjecture that optimizing this distribution will further increase the advantage of the QPT approach in our problems. We leave it to future work to develop a methodology to identify more optimal distributions.

Furthermore, our choice of swap probability (Eq. (11)) was based on an intuitive argument, and we provided examples of other possible choices in Appendix D. It would be ideal to identify a choice that can satisfy some appropriate notion of detailed balance. We believe that this is the most important open theoretical problem with our proposal. Such a formalism would also allow us to adopt some of the strategies and methods using in conjunction with QPT to characterize the hardness of the landscape, such as using the autocorrelation time of the replica random walk as a proxy for the mixing time [23].

# 5 Code

Code for this project can be found at https://github.com/talbash/QPTforRBM.

# Acknowledgements

We would like to thank the UNM Center for Advanced Research Computing, supported in part by the National Science Foundation, for providing the high performance computing resources used in this work.

**Funding information**    This material is based upon work supported by the National Science Foundation under Grant No. 2037755. This material is based upon work supported by the Air Force Office of Scientific Research under award number FA9550-22-1-0498. Any opinions, findings, and conclusions or recommendations expressed in this material are those of the author(s) and do not necessarily reflect the views of the United States Air Force. CS and QC were supported by Sandia National Laboratories' Laboratory Directed Research and Development program (Project 222396). Sandia National Laboratories is a multi-mission laboratory managed and operated by National Technology and Engineering Solutions of Sandia, LLC, a wholly owned subsidiary of Honeywell International, Inc., for DOE's National Nuclear Security Administration under contract DE-NA0003525. This paper describes objective technical results and analysis. Any subjective views or opinions that might be expressed in the paper do not necessarily represent the views of the U.S. Department of Energy or the United States Government.

# A    Review of Stochastic Reconfiguration

We consider a (possibly unnormalized) wave function $|\psi(\alpha)\rangle$ parameterized by $K$ complex parameters denoted by $\alpha = (\alpha_1, \dots, \alpha_K)^T$. We are interested in finding the parameter values $\alpha$ that minimize

$$E(\alpha) \equiv \frac{\langle\psi(\alpha)|H|\psi(\alpha)\rangle}{\langle\psi(\alpha)|\psi(\alpha)\rangle} = \langle H\rangle. \tag{A.1}$$

The true minimum is given by the ground state energy $E_{\text{GS}}$. Because the parameterization of the wave function $|\psi(\alpha)\rangle$ is unlikely to be general enough to capture the ground state *exactly*, we can only hope for $|\psi(\alpha)\rangle$ (when normalized) to provide an approximation to the ground state.

Assume the wave function has an expansion in a basis of the form: $|\psi(\alpha)\rangle = \sum_x \psi_x(\alpha)|x\rangle$. For simplicity, we will assume that $\psi_x(\alpha)$ only depends on $\alpha_k$ and not on both $(\alpha_k, \alpha_k^*)$. Let us consider a small variation of the parameters $\alpha'_k = \alpha_k + \delta\alpha_k$ Let us expand the wave function:

$$|\psi(\alpha')\rangle = |\psi(\alpha)\rangle + \sum_k \delta\alpha_k \sum_x \left(\frac{1}{\psi_x(\alpha)} \frac{\partial}{\partial\alpha_k} \psi_x(\alpha)\right) \psi_x(\alpha)|x\rangle$$
$$+ \frac{1}{2} \sum_{k,k'} \delta\alpha_k \delta\alpha_{k'} \left(\frac{1}{\psi_x(\alpha)} \frac{\partial}{\partial\alpha_k} \frac{\partial}{\partial\alpha_{k'}} \psi_x(\alpha)\right) \psi_x(\alpha)|x\rangle + \dots \tag{A.2}$$

It is convenient to define the operators:

$$O_k(\alpha) = \sum_x \frac{1}{\psi_x(\alpha)} \frac{\partial}{\partial\alpha_k} \psi_x(\alpha)|x\rangle\langle x|, \tag{A.3 a}$$

$$M_{kk'}(\alpha) = \sum_x \frac{1}{\psi_x(\alpha)} \frac{\partial}{\partial\alpha_k} \frac{\partial}{\partial\alpha_{k'}} \psi_x(\alpha)|x\rangle\langle x|, \tag{A.3 b}$$

(We will suppress their $\alpha$ dependence for notational brevity) such that we can write

$$|\psi(\alpha')\rangle = |\psi(\alpha)\rangle + \sum_k \delta\alpha_k O_k |\psi(\alpha)\rangle + \frac{1}{2} \sum_{k,k'} \delta\alpha_k \delta\alpha_{k'} M_{kk'} |\psi(\alpha)\rangle + \dots$$
$$= \left(1 + \sum_k \delta\alpha_k \langle O_k\rangle + \dots\right) |\psi(\alpha)\rangle + \sum_k \delta\alpha_k (O_k - \langle O_k\rangle) |\psi(\alpha)\rangle$$
$$+ \frac{1}{2} \sum_{k,k'} \delta\alpha_k \delta\alpha_{k'} (M_{kk'} - \langle M_{kk'}\rangle) |\psi(\alpha)\rangle + \dots, \tag{A.4}$$

where

$$\langle O_k(\alpha)\rangle = \frac{\langle\psi(\alpha)|O_k(\alpha)|\psi(\alpha)\rangle}{\langle\psi(\alpha)|\psi(\alpha)\rangle}, \tag{A.5 a}$$

$$\langle M_{kk'}(\alpha)\rangle = \frac{\langle\psi(\alpha)|M_{kk'}(\alpha)|\psi(\alpha)\rangle}{\langle\psi(\alpha)|\psi(\alpha)\rangle}. \tag{A.5 b}$$

We will denote the coefficient of the $|\psi(\alpha)\rangle$ term in Eq. (A.4) by $\delta\alpha_0$, such that $\langle\psi(\alpha)|\psi(\alpha')\rangle = \delta\alpha_0\langle\psi(\alpha)|\psi(\alpha)\rangle$ and

$$\langle\psi(\alpha')|\psi(\alpha')\rangle = |\delta\alpha_0|^2 \langle\psi(\alpha)|\psi(\alpha)\rangle$$
$$+ \sum_{k,k'} \delta\alpha_k^* \delta\alpha_{k'} \langle\psi(\alpha)| \left(O_k^\dagger - \langle O_k^\dagger\rangle\right)(O_{k'} - \langle O_{k'}\rangle) |\psi(\alpha)\rangle + \dots$$
$$= |\delta\alpha_0|^2 \left(1 + \frac{1}{|\delta\alpha_0|^2} \sum_{k,k'} \delta\alpha_k^* \delta\alpha_{k'} S_{k,k'} + \dots\right) \times \langle\psi(\alpha)|\psi(\alpha)\rangle, \tag{A.6}$$

where we have defined the covariance matrix

$$S_{kk'} = \frac{\langle\psi(\alpha)| \left(O_k^\dagger - \langle O_k^\dagger\rangle\right)(O_{k'} - \langle O_{k'}\rangle) |\psi(\alpha)\rangle}{\langle\psi(\alpha)|\psi(\alpha)\rangle}$$
$$= \langle O_k^\dagger O_{k'}\rangle - \langle O_k^\dagger\rangle\langle O_{k'}\rangle. \tag{A.7}$$

Here we have suppressed the dependence of $S$ on $\alpha$. We see from Eq. (A.6) that renormalizing $|\psi(\alpha')\rangle$ by $\delta\alpha_0$ amounts to rescaling $\delta\alpha_k$ by $\delta\alpha_0$. We will exploit this freedom later. Finally,

for completeness, we note that we can expand the energy as:

$$E(\alpha') = E(\alpha) + \frac{1}{\langle\psi(\alpha)|\psi(\alpha)\rangle}\sum_k \left(\delta\alpha_k^* \langle\psi(\alpha)|\left(O_k^\dagger - \langle O_k^\dagger\rangle\right)H|\psi(\alpha)\rangle\right.$$

$$\left. + \delta\alpha_k\langle\psi(\alpha)|H\left(O_k - \langle O_k\rangle\right)|\psi(\alpha)\rangle\right) + \dots$$

$$= E(\alpha) + \sum_k \left(\delta\alpha_k^* F_k + \delta\alpha_k F_k^*\right) + \dots, \tag{A.8}$$

where we have defined

$$F_k = \frac{\langle\psi(\alpha)|\left(O_k^\dagger - \langle O_k^\dagger\rangle\right)(H - E(\alpha))|\psi(\alpha)\rangle}{\langle\psi(\alpha)|\psi(\alpha)\rangle}$$

$$= \langle O_k^\dagger H\rangle - \langle O_k^\dagger\rangle\langle H\rangle. \tag{A.9}$$

Here we have suppressed the dependence of $F$ on $\alpha$.

For the SR algorithm we further demand that:

$$|\psi(\alpha')\rangle = P(\Lambda - H)|\psi(\alpha)\rangle, \tag{A.10}$$

where $\Lambda$ is taken to be a sufficiently large (larger than the largest eigenvalue of $H$) positive number and $P$ is the projector onto the subspace of the parameterization. By acting with $\langle\psi(\alpha)|$ and $\langle\psi(\alpha)|\left(O_k^\dagger - \langle O_k^\dagger\rangle\right)$, and by restricting to linear order in $\delta\alpha$, we get $K+1$ linear equations:

$$\delta\alpha_0 = \Lambda - E(\alpha), \tag{A.11 a}$$

$$\sum_{k'}\delta\alpha_{k'}\langle\psi(\alpha)|\left(O_k^\dagger - \langle O_k^\dagger\rangle\right)(O_{k'} - \langle O_{k'}\rangle)|\psi(\alpha)\rangle = -\langle\psi(\alpha)|\left(O_k^\dagger - \langle O_k^\dagger\rangle\right)H|\psi(\alpha)\rangle$$

$$= -\langle\psi(\alpha)|\left(O_k^\dagger - \langle O_k^\dagger\rangle\right)(H - E(\alpha))|\psi(\alpha)\rangle, \tag{A.11 b}$$

we get the linear equation:

$$-F_k = \sum_{k'} S_{kk'}\delta\alpha_{k'} \Rightarrow \delta\alpha_k = -\sum_{k'} S_{kk'}^{-1} F_{k'}. \tag{A.12}$$

Since $S$ may not be invertible, $S^{-1}$ is strictly speaking the Moore-Penrose pseudoinverse. Therefore, the update rule for the parameters $\alpha$ is given by:

$$\alpha_k' = \alpha_k + \delta\alpha_k = \alpha_k - \sum_{k'} S_{kk'}^{-1} F_{k'}. \tag{A.13}$$

Furthermore, if we renormalize the state $\psi(\alpha')$ by $\delta\alpha_0$, we effectively are rescaling $\delta\alpha_k$ by $\delta\alpha_0$. Therefore, our update rule after this renormalization is given by:

$$\alpha_k' = \alpha_k + \frac{\delta\alpha_k}{\delta\alpha_0} = \alpha_k - \frac{1}{\delta\alpha_0}\sum_{k'} S_{kk'}^{-1} F_{k'}, \tag{A.14}$$

and we have (from Eq. (A.6)):

$$\frac{\langle\psi(\alpha')|\psi(\alpha')\rangle - \langle\psi(\alpha)|\psi(\alpha)\rangle}{\langle\psi(\alpha)|\psi(\alpha)\rangle} = \frac{1}{|\delta\alpha_0|^2}\sum_{k,k'}\delta\alpha_k^* s_{k,k'}\delta\alpha_{k'} + \dots \tag{A.15}$$

By repeating this iterative scheme, convergence is reached when the ratio $\delta\alpha_k/\delta\alpha_0 \to 0$: we have from Eq. (A.8 ):

$$E(\alpha') - E(\alpha) = -\frac{1}{|\delta\alpha_0|^2}\sum_{k,k'}\left(F_k^* S_{kk'}^{-1} F_{k'} + \left(F_k^* S_{kk'}^{-1} F_{k'}\right)^*\right). \tag{A.16}$$

Because the covariance matrix is a positive semi-definite matrix, the term in the sum is positive, which says that on every iteration the energy decreases, and in the limit of $\delta\alpha_k/\delta\alpha_0 \to 0$, the energy ceases to change.

As we noted earlier, there is some freedom in picking the value of $\Lambda$. Since $\Lambda$ controls the value of $\delta\alpha_k$, which in turn controls both the parameter update rate (Eq. (A.14 )) and the wave function change rate (Eq. (A.15 )), in practice we treat $\delta\alpha_0^{-1} \equiv \gamma$ as a free parameter that is chosen to get a stable convergence. The value of $\gamma$ is the learning rate. As we describe in Appendix B, we will use an adaptive scheme to update this learning rate.

In the case of the RBM ansatz, the operators $O_k$ can be calculated analytically. The relevant quantities are given by:

$$\frac{1}{\psi_x(\alpha)}\frac{\partial}{\partial a_i}\psi_x(\alpha) = (1 - 2x_i), \tag{A.17 a}$$

$$\frac{1}{\psi_x(\alpha)}\frac{\partial}{\partial b_\mu}\psi_x(\alpha) = \tanh\left(\theta(x)\right), \tag{A.17 b}$$

$$\frac{1}{\psi_x(\alpha)}\frac{\partial}{\partial W_{i\mu}}\psi_x(\alpha) = (1 - 2x_i)\tanh\left(\theta(x)\right). \tag{A.17 c}$$

# B  Adaptive Learning Rate Scheme

We begin by interpreting Eq. (3) as being a discretization of a dynamical equation for $\alpha$ with $\gamma$ being the step size; therefore, let us recast it as:

$$\frac{d}{d\gamma}\alpha_k(\gamma) = -\sum_{k'}(S(\alpha(\gamma))^{-1})_{kk'}F_{k'}(\alpha(\gamma)) \equiv f_k(\alpha(\gamma)), \tag{B.1}$$

where we have introduced the dependence of $\alpha$ on $\gamma$. In this form, we can use a Heun second order consistent integrator to adaptively update $\gamma$. We calculate two quantities:

$$\vec{\kappa}_1 = \vec{f}(\vec{\alpha}(\gamma)), \tag{B.2 a}$$

$$\vec{\kappa}_2 = \vec{f}(\vec{\alpha}(\gamma) + \Delta\gamma\vec{\kappa}_1), \tag{B.2 b}$$

and then update using:

$$\vec{\alpha}(\gamma + \Delta\gamma) = \vec{\alpha}(\gamma) + \frac{\Delta\gamma}{2}(\vec{\kappa}_1 + \vec{\kappa}_2). \tag{B.3}$$

In order to dynamically adjust the step $\Delta\gamma$, we calculate the ANN parameters at the same time $\gamma + \Delta\gamma$ but now using two $\Delta\gamma/2$ steps. Therefore, we define:

$$\vec{\kappa}_3 = \vec{f}\left(\vec{\alpha}(\gamma) + \frac{\Delta\gamma}{2}\vec{\kappa}_1\right), \tag{B.4 a}$$

$$\vec{\kappa}_4 = \vec{f}\left(\vec{\alpha}(\gamma) + \frac{\Delta\gamma}{4}(\vec{\kappa}_1 + \vec{\kappa}_3)\right), \tag{B.4 b}$$

$$\vec{\kappa}_5 = \vec{f}\left(\vec{\alpha}(\gamma) + \frac{\Delta\gamma}{4}(\vec{\kappa}_1 + \vec{\kappa}_3) + \frac{\Delta\gamma}{2}\vec{\kappa}_4\right), \tag{B.4 c}$$

such that

$$\vec{\alpha}'(\gamma + \Delta\gamma) = \vec{\alpha}(\gamma) + \frac{\Delta\gamma}{4}\left(\vec{\kappa}_1 + \vec{\kappa}_3\right) + \frac{\Delta\gamma}{4}\left(\vec{\kappa}_4 + \vec{\kappa}_5\right). \tag{B.5}$$

We define $\Delta\alpha = \|\vec{\alpha}'(\gamma + \Delta\gamma) - \vec{\alpha}(\gamma + \Delta\gamma)\|_{S(\alpha(\gamma))}$, where the norm here is defined as:

$$\|\vec{x}\|_{S(\alpha(\gamma))} = \frac{1}{M}\sqrt{\sum_{k,k'=1}^{M} x_k^* S_{kk'}(\alpha(\gamma)) x_{k'}}. \tag{B.6}$$

The adjusted time step is then taken to be [17]:

$$\Delta\gamma' = \Delta\gamma\left(\frac{6\epsilon}{\Delta\alpha}\right)^{1/3}, \tag{B.7}$$

for a fixed tolerance $\epsilon$. In our simulations, we fix $\epsilon = 10^{-6}$. We note that in a single update step in this framework requires performing 5 different calculations of $S$ and $F$, one for each of the different $\kappa$ values.

## C  Calculating the expectation value of operators with respect to an RBM ansatz

We begin by first considering a diagonal operator $D$, meaning it is diagonal in the computational basis. In this case, the expectation value of the operator can be straightforwardly calculated using the RBM ansatz in Eq. (1) (we will suppress the $\alpha$ dependence):

$$\langle D \rangle = \frac{1}{\mathcal{Z}}\sum_x |\psi_x|^2 \langle x|D|x\rangle, \tag{C.1}$$

where $\mathcal{Z} = \langle\psi|\psi\rangle = \sum_x |\psi_x|^2$. Note that $|\psi\rangle/\sqrt{\mathcal{Z}}$ defines a normalized state.

Let us now consider the expectation value of an arbitrary Pauli operator acting on $n$ qubits, which we denote by $P$. We can express the expectation value as:

$$\begin{aligned}
\langle P \rangle &= \frac{1}{\mathcal{Z}}\sum_{x,x'} \psi_x^* \psi_{x'} \langle x|P|x'\rangle \\
&= \frac{1}{\mathcal{Z}}\sum_x |\psi_x|^2 \left(\sum_{x'} \frac{\psi_{x'}}{\psi_x}\langle x|P|x'\rangle\right),
\end{aligned} \tag{C.2}$$

where the term in parenthesis defines the local Pauli operator $P(x)$:

$$P(x) = \sum_{x'} \frac{\psi_{x'}}{\psi_x}\langle x|P|x'\rangle. \tag{C.3}$$

For a given $P$, let $\mathcal{I}_\alpha$, $\alpha = x, y, z$, denote the qubit indices where a single-qubit Pauli operator $\sigma^\alpha$ acts. For example, consider the Pauli operator acting on 3 qubits $P = \sigma_2^x \otimes \sigma_1^y \otimes \sigma_0^z$; in this case we would have $\mathcal{I}_x = \{2\}$, $\mathcal{I}_y = \{1\}$, $\mathcal{I}_z = \{0\}$. For the RBM ansatz in Eq. (1), we can calculate the local Pauli operator straightforwardly:

$$\begin{aligned}
P(x) = &\left[\prod_{j\in\mathcal{I}_z}(-1)^{x_j}\right]\left[i^{|\mathcal{I}_y|}\prod_{j\in\mathcal{I}_y}(-1)^{1-x_j}\right]\exp\left(-2\sum_{j\in\mathcal{I}_x\cup\mathcal{I}_y}a_j(1-2x_j)\right) \\
&\times\prod_{\mu=1}^{m}\frac{\cosh\left(\theta_\mu(x) - 2\sum_{j\in\mathcal{I}_x\cup\mathcal{I}_y}W_{j\mu}(1-2x_j)\right)}{\cosh\left(\theta_\mu(x)\right)},
\end{aligned} \tag{C.4}$$

where $\theta_\mu(x) = b_\mu + \sum_{j=1}^{n} W_{j\mu}(1 - 2x_j)$.

The above prescription allows us to calculate the expectation values in the covariance matrix and force vector in Eq. (4). For example, let us consider the term $\langle O_k^\dagger H \rangle$. We can write:

$$
\begin{aligned}
\langle O_k^\dagger H \rangle &= \frac{1}{\mathcal{Z}} \sum_{x,x'} \psi_x^* \psi_{x'} \langle x | O_k^\dagger H | x' \rangle \\
&= \frac{1}{\mathcal{Z}} \sum_{x,x',x''} \psi_x^* \psi_{x'} \langle x | O_k^\dagger | x'' \rangle \langle x'' | H | x' \rangle \\
&= \frac{1}{\mathcal{Z}} \sum_x |\psi_x|^2 \left( \sum_{x''} \frac{\psi_{x''}}{\psi_x} \langle x | O_k^\dagger | x'' \rangle \times \sum_{x'} \frac{\psi_{x'}}{\psi_{x''}} \langle x'' | H | x' \rangle \right) \\
&= \frac{1}{\mathcal{Z}} \sum_x |\psi_x|^2 \langle x | O_k^\dagger | x \rangle H(x),
\end{aligned}
\tag{C.5}
$$

where we have used that the operators $O_k$ are diagonal and where we defined a local Hamiltonian:

$$
H(x) = \sum_{x'} \frac{\psi_{x'}}{\psi_x} \langle x | H | x' \rangle.
\tag{C.6}
$$

Since any Hamiltonian can be expressed in terms of diagonal operators and arbitrary Pauli operators, we can calculate the local Hamiltonian using our prescription above for diagonal and local Pauli operators.

In the case of the completely symmetric RBM ansatz, the expectation values take a simple form. For example, we can express the contribution of the transverse field as:

$$
\begin{aligned}
\sum_{i=1}^{n} \langle \sigma_i^x \rangle = \sum_{w=0}^{n} \binom{n}{w} |\psi_w(\alpha)|^2 \times \bigg( & n\delta_{w0} \frac{\psi_{w+1}(\alpha)}{\psi_w(\alpha)} + n\delta_{wn} \frac{\psi_{w-1}(\alpha)}{\psi_w(\alpha)} \\
& + \delta_{0<w<n} \left( (n-w) \frac{\psi_{w+1}(\alpha)}{\psi_w(\alpha)} + w \frac{\psi_{w-1}(\alpha)}{\psi_w(\alpha)} \right) \bigg),
\end{aligned}
\tag{C.7}
$$

where we have defined $\psi_w(\alpha)$ as

$$
\psi_w(\alpha) = e^{a(n-2w)} \prod_{\mu=1}^{m} \cosh\left( b_\mu + W_\mu(n-2w) \right).
\tag{C.8}
$$

We can also calculate:

$$
\begin{aligned}
\sum_{i \neq j}^{n} \langle \sigma_i^x \sigma_j^x \rangle = \sum_{w=0}^{n} \binom{n}{w} |\psi_w(\alpha)|^2 \times \bigg[ & \delta_{w0} n(n-1) \frac{\psi_{w+2}}{\psi_w} + \delta_{wn} n(n-1) \frac{\psi_{w-2}}{\psi_w} \\
& + \delta_{w1} \left( 2(n-1) + (n-1)(n-2) \frac{\psi_{w+2}}{\psi_w} \right) \\
& + \delta_{w,n-1} \left( 2(n-1) + (n-1)(n-2) \frac{\psi_{w-2}}{\psi_w} \right) \\
& + \delta_{1<w<n-1} \left( 2w(n-w) + (n-w)(n-w-1) \frac{\psi_{w+2}}{\psi_w} \right. \\
& \left. + w(w-1) \frac{\psi_{w-2}}{\psi_w} \right) \bigg].
\end{aligned}
\tag{C.9}
$$

# D Alternative Parallel Tempering Schemes

## D.1 Standard deviation of $H^2$ as a measure of temperature

As we indicated in the main text, there is a lot of freedom in choosing the training Hamiltonian and the parallel tempering update scheme. Here we present an alternative approach. We can choose the Hamiltonian for the $r$th replica to be given by

$$H(\Gamma_r) = (1 - \Gamma_r)H_M + \Gamma_r H_T \,, \tag{D.1}$$

where $H_T$ is the target Hamiltonian whose ground state we are actually after and $H_M$ is a suitably chosen mixer Hamiltonian with a trivial ground state. For example, in the absence of any constraints, the mixer can be taken to be the uniform transverse field Hamiltonian, $H_M = -H_0 \sum_{i=1}^{n} \sigma_i^x$, where $H_0$ is a characteristic energy scale of $H_T$. In this case, for replicas with $\Gamma_r$ close to 0 the training Hamiltonian is dominated by the transverse field, and the ground state is the uniform superposition state. This is analogous to the high-temperature limit in "standard" PT, where the different replicas operate at different temperatures but with the same Hamiltonian. For replicas with $\Gamma_r$ close to 1, the training Hamiltonian is dominated by the target Hamiltonian, and the ground state is close to the target ground state. This is analogous to the low-temperature limit in standard parallel tempering. The parameter $\Gamma_r$ thus allows us to interpolate between the target parameter landscape at $\Gamma_r = 1$ and a much simpler parameter landscape at $\Gamma_r = 0$. For simplicity we choose our distribution of $\Gamma_r$ to be linearly spaced,

$$\Gamma_r = 1 - \frac{r}{N-1} \,, \quad r = 0, \dots, N-1 \,. \tag{D.2}$$

We then propose the following PT update. Since $\Gamma_r$ controls the relative energy scales of $H_T$ and $H_M$, it is not a suitable candidate for the temperature as it was in the main text, so we propose a different choice. We expect the standard deviation $\sigma_r$ to be 0 for $\Gamma_r = 1$ ($r = 0$) because we expect the ANN to reach a good approximation of an eigenstate of $H_T$, assuming the number ANN parameters is sufficiently large. Similarly, for larger $r$ replicas this quantity should be non-zero since we expect the ANN to be a good approximation of an eigenstate of $H(\Gamma_r)$. Therefore, we can think of $\sigma_r$ as being a measure of the broadening of the eigenstates of $H_T$ due to the "noise" introduced by $\Gamma_r < 1$. In this sense, we can think of $\sigma_r$ as being a kind of temperature. We thus propose the following probability rule for swap updates between neighboring pairs $r$ and $r+1$:

$$p_{r \leftrightarrow r+1} = \begin{cases} 1, & \text{if } r = 0 \text{ and } \overline{E_{r+1}} < \overline{E_r} \,, \\ 0, & \text{if } r = 0 \text{ and } \overline{E_{r+1}} > \overline{E_r} \,, \\ \min\left(1, \exp\left[\left(\overline{E_r} - \overline{E_{r+1}}\right)\left(\frac{2}{\overline{\sigma_r}} - \frac{2}{\overline{\sigma_{r+1}}}\right)\right]\right), & \text{otherwise} \,. \end{cases} \tag{D.3}$$

While it is not strictly necessary, the $r = 0$ replica only swaps its configuration with its neighbor if its neighbor has a better approximation of the ground state. This is equivalent to what happens in the main text at zero temperature. The factor of 2 in $2/\overline{\sigma_r}$ is a choice we make to ensure that enough replica swaps happen in our simulations. This is a hyper-parameter that could be optimized.

We show the QPT simulation results using this rule for the Precipice problem in Fig. 7 and for the $H_4$ Rectangle in Fig. 8. The results are comparable to our results from the main text (Figs. 4, 5, and 6), although the performance using the rule in the main text is better. We also emphasize that our simulations with this rule did not succeed at finding a good approximation of the ground state with $N = 5$ replicas, while the approach in the main text does.

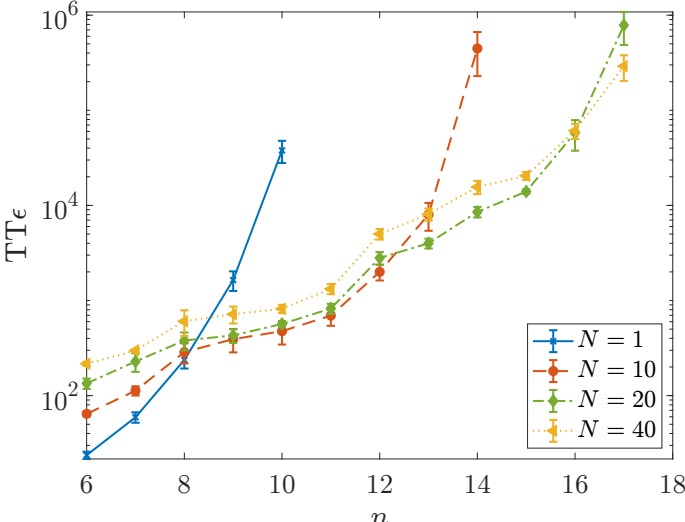

Figure 7: Optimum time to epsilon to reach a relative error of $\epsilon = 10^{-1}$ for the Precipice problem as a function of system size $n$ using the QPT swap probability rule in Eq. (D.3 ) with different number of QPT replicas $N$ with $N_{\text{max}} = 40$. The error bars correspond to the 95% confidence interval calculated using a bootstrap with $10^3$ resamplings of the data.

## D.2 Almost constant swap probability

In what follows we use the same the same definition of the training Hamiltonian as in the main text (Eq. (7)), as well as the same distribution of $\Gamma_r$ values (Eq. (8)). We consider a swap rule that abandons the idea of a temperature altogether; instead we consider the following swap probability rule:

$$p_{r\leftrightarrow r+1} = \begin{cases} 1, & \text{if } r = 0 \text{ and } \overline{E_{r+1}} < \overline{E_r}, \\ 0, & \text{if } r = 0 \text{ and } \overline{E_{r+1}} > \overline{E_r}, \\ 1/4, & \text{otherwise}. \end{cases} \tag{D.4}$$

The motivation for this choice is as follows. The $r = 0$ replica only swaps its configuration with its neighbor if its neighbor has a better approximation of the ground state. This is equivalent to what happens in the main text at zero temperature. For the remaining replicas, we swap the replicas with probability 1/4 irrespective of the energies of the configurations. We show the QPT simulation results using this rule for the Precipice problem in Fig. 9 and for the $H_4$ Rectangle in Fig. 10. The results are comparable to our results from the main text (Figs. 4, 5, and 6), although the performance using the rule in the main text is better in all cases, although there are some qualitative differences. For the Precipice problem, we see a clear separation between the different $N$ values for intermediate $n$ values, which suggests that the constant swap probability with a smaller separation between replicas is hindering performance compared to the probability rule in the main text. For the $H_4$ Rectangle, the larger error bars suggest less consistency in the performance using this scheme.

## E   Short Note On the Bootstrap Procedure

Let us assume we have repeated our stochastic simulations $n$ times and gotten the set of outcomes $\{x_1, x_2, \ldots, x_n\}$. The outcome of the $i$-th experiment can be treated as a random variable $\mathbf{X_i}$, with the $i$-th experimental outcome given $\mathbf{X_i} = x_i$. We will assume that all the random vari-

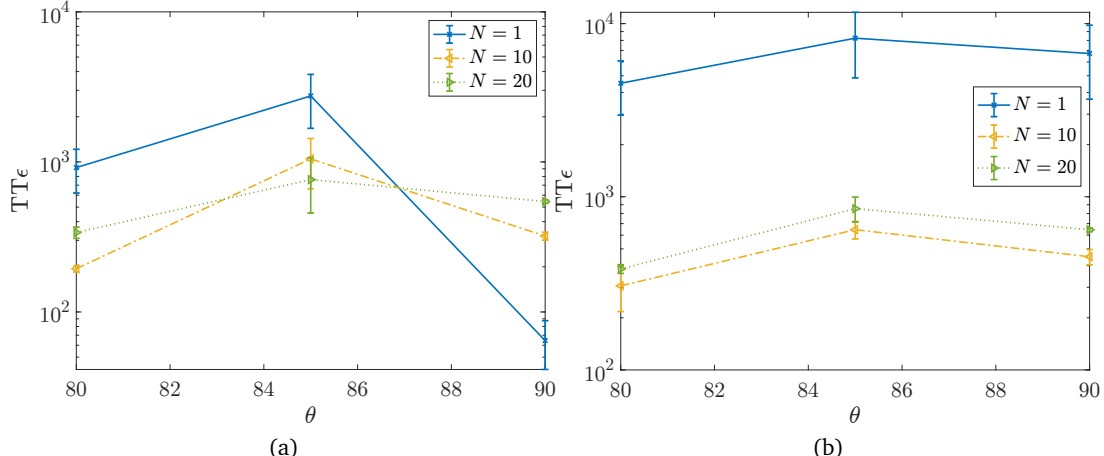

Figure 8: Optimum time to epsilon to reach a relative error of $\epsilon = 10^{-3}$ for the H$_4$ Rectangle at different angles with (a) fixed Hamming weight exact sampling and (b) full exact sampling using the QPT swap probability rule in Eq. (D.3). We use $N_{\max} = 20$. The error bars correspond to the 95% confidence interval calculated using a bootstrap with $10^3$ resamplings of the data. The lines are to guide the eye.

ables are independent and identically distributed with mean $\mu_x$ and standard deviation $\sigma_x$. Our goal is to report the mean and uncertainty of some property of the random variables. For example, if we are interested in estimating $\mu_x$, we may use the the sample mean $\bar{\mathbf{X}}$ as our function:

$$\bar{\mathbf{X}} = \frac{1}{n} \sum_{i=1}^{n} \mathbf{X_i}. \tag{E.5}$$

From the Central Limit Theorem, we expect that $\bar{\mathbf{X}} \sim \mathcal{N}(\mu_x, \sigma_x^2/n)$ for sufficiently large $n$. So for sufficiently large $n$, we can report our 95% confidence estimate of $\mu_x$ as $\bar{x} \pm 2\frac{\sigma_x}{\sqrt{n}}$ where $\bar{x} = \frac{1}{n} \sum_{i=1}^{n} x_i$. We can estimate $\sigma_x^2$ by using the sample variance for example.

However, if instead of estimating $\mu_x$, we were interested in the median of $\mathbf{X_i}$, then it is not obvious how to use the above method to give confidence estimates. The bootstrap method allows us to mitigate this problem in a simple way.

Bootstrapping requires resampling (with replacement) the data $(x_1, \ldots, x_n)$ uniformly to acquire $n_b$ bootstrap resamples. Each bootstrap resample $S_i$ picks $n$ of the data outcomes randomly with replacement, i.e. $S_i = \{x_{i_1}, x_{i_2}, \ldots x_{i_n}\}$. So a bootstrap resample effectively amounts to randomly picking indices from 1 to $n$ with replacement. For each $S_i$, we calculate the property we are interested in, like the median, giving a value $F_i$. We can treat the set $\{F_1, F_2, \ldots, F_{n_b}\}$ as $n_b$ samples of the random variables $\mathbf{F_i}$. We now calculate the sample mean of $\mathbf{F}$:

$$\bar{\mathbf{F}} = \frac{1}{n_b} \sum_{i=1}^{n_b} \mathbf{F_i}. \tag{E.6}$$

Now we can use the Central Limit Theorem (for sufficiently large $n_b$) that $\bar{\mathbf{F}} \sim \mathcal{N}$. We can now report our estimate with 95% confidence interval as:

$$\bar{F} \pm 2\sigma_F, \tag{E.7}$$

where $\bar{F}$ is the sample mean:

$$\bar{F} = \frac{1}{n_b} \sum_{i=1}^{n_b} F_i, \tag{E.8}$$

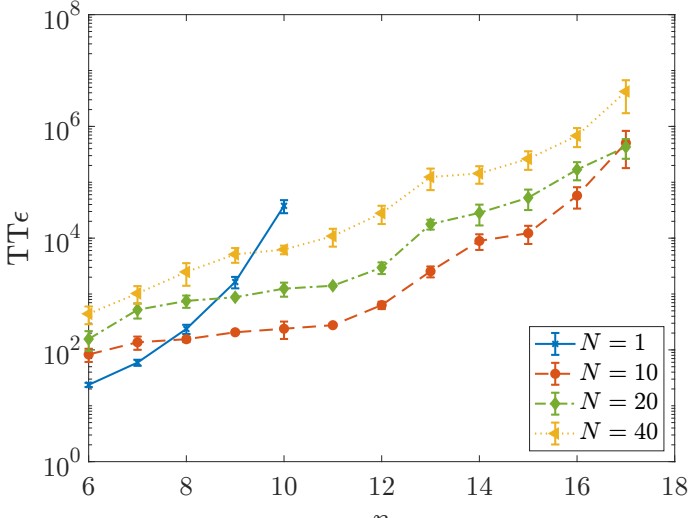

Figure 9: Optimum time to epsilon to reach a relative error of $\epsilon = 10^{-1}$ for the Precipice problem as a function of system size $n$ using the QPT swap probability rule in Eq. (D.4) with different number of QPT replicas $N$ with $N_{\max} = 40$. The error bars correspond to the 95% confidence interval calculated using a bootstrap with $10^3$ resamplings of the data.

and $\sigma_F$ is the standard deviation:

$$\sigma_F^2 = \frac{1}{n_b - 1} \sum_{i=1}^{n_b} (F_i - \bar{F})^2 \,. \tag{E.9}$$

# F  The $H_4$ rectangle

The $H_4$ rectangle, illustrated in Fig. 11, consists of four hydrogen atoms placed in a rectangle, each with a distance to the center of $R$. The separation between these atoms can be further controlled by the angle between adjacent hydrogen atoms $\theta$. When $\theta$ is far away from 90°, the system resembles two disconnected $H_2$ molecules and the energy can be approximately calculated as twice the energy of an equivalent $H_2$ molecule. As the angle approaches 90°, however, the distance between the different hydrogen atoms becomes equivalent and the system becomes degenerate, leading to a frustrated system with a peak in energy at 90°. By moving from a rectangular system to a square one, the system can be brought arbitrarily close to a strongly correlated, two configuration system. This configuration has been shown to lead to difficulty for many atomic modeling algorithms, notably including Coupled Cluster methods, which incorrectly predict a cusp and minimum at 90° for certain radii [51]. Throughout this work, following Pfau *et al.* and others [54, 55], we have used a radius of 1.738 Å, which places the $H_4$ rectangle in a regime where the atoms are nearing full dissociation. Coupled Cluster methods have been shown to fail at this radius [51], and single replica RBM runs often struggle to find the ground state energy as seen in the main text, providing a useful benchmark for our parallel tempering scheme.

To create inputs for our calculations, we consider the Fermionic problem in the second quantitized formalism

$$H = \sum_{i,j} t_{ij} c_i^\dagger c_j + \sum_{i,j,k,m} u_{ijkm} c_i^\dagger c_k^\dagger c_m c_j \,, \tag{F.1}$$

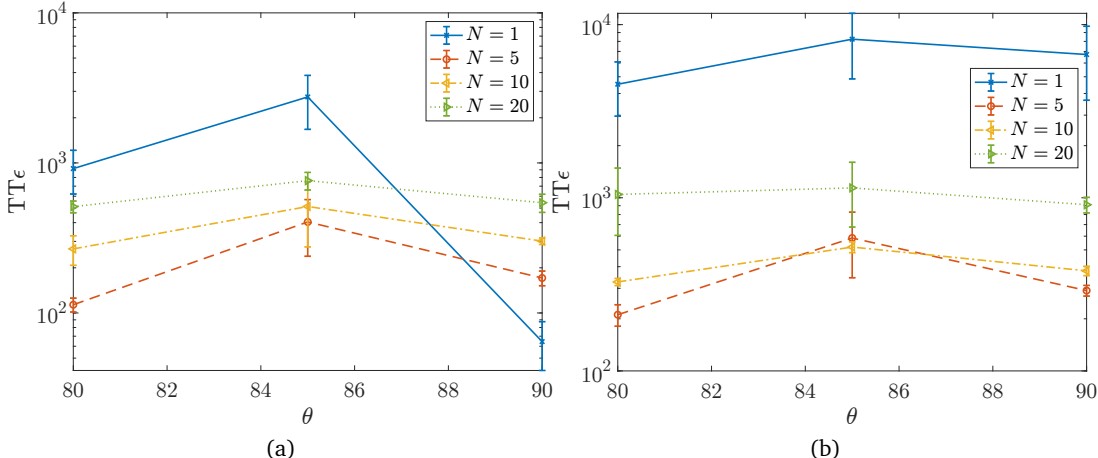

Figure 10: Optimum time to epsilon to reach a relative error of $\epsilon = 10^{-3}$ for the $H_4$ Rectangle at different angles with (a) fixed Hamming weight exact sampling and (b) full exact sampling using the QPT swap probability rule in Eq. (D.4). We use $N_{\max} = 20$. The error bars correspond to the 95% confidence interval calculated using a bootstrap with $10^3$ resamplings of the data. The lines are to guide the eye.

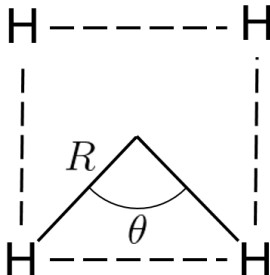

Figure 11: The arrangement of hydrogen atoms in the $H_4$ rectangle. The configuration is controlled by the radius $R$ and the angle $\theta$.

where we define the fermionic annihilation and creation operators with the anticommutation relation $\{c_i^\dagger, c_j\} = \delta_{i,j}$. We define $t_{ij}$ and $u_{ijkm}$ as one- and two-body integrals, respectively. We then map this fermionic Hamiltonian onto a spin Hamiltonian with the form

$$H = \sum_{j=1}^{r} h_j \sigma_j \,, \tag{F.2}$$

where $h_j$ are coefficients and $\sigma_j$ is an N-fold tensor product of single-qubit Pauli operators $I$, $\sigma^x$, $\sigma^y$, and $\sigma^z$. We use the canonical Jordan-Wigner transformation [26] to map Hamiltonians of the form of Eq. F.1 to Eq. F.2. We use a minimal STO-3G basis set [56] and the OpenFermion software package [57] to generate and map Hamiltonians for the $H_4$ problem.

For $\theta = 80°$, the ground state energy of the 8 qubit Hamiltonian is $-1.88016686$; for $\theta = 85°$, the ground state energy is $-1.87584118$; for $\theta = 90°$, the ground state energy is $-1.87420093$.

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
