# Peer review of "Quantum-Inspired Tempering for Ground State Approximation using Artificial Neural Networks"

_SciPost Physics, doi:SciPost Phys. 14, 121 (2023)_

## Round 2 · Referee Report · Pranay Patil (Referee 1) · 2022-12-18

Strengths

1-Addresses an important aspect of the variational approach to quantum many-body systems. The topic discussed is of broad interest for adiabatic quantum computing and statistical mechanics communities.

2-The method developed is general and can be applied to a wide variety of optimization problems, and thus has a significant scope of application.

3-The paper is presented in a pedagogical way, with a clear flow of content, and appropriate relegation of details and derivations to appendices. The overarching theme is easy to understand.

Weaknesses

1-Given that the parallel tempering apparatus is the heart of the method, detailed information about its performance is required. This has not been included by the authors. A suggestion for this would be studies of $E_r(t)$ and $\sigma_r(t)$ throughout the training period. Additional discussions about the performance of the training for all replicas is also beneficial, as it helps provide an intuition of the optimization process.

2-The text below Eq.10 is lacking in clarity (I believe you mean "over 10 updates of SR by $\bar{O}$" and not $\langle\bar{O}\rangle$). The presentation here would benefit from a discussion about the performance when using just $O$, instead of the average. This also helps with a better understanding of Eq.12 .

3-The discussion below Eq.14 would benefit from an example illustrating the procedure of calculating the probabilities associated with $u$. This is not completely necessary as it may be out of the scope of this work, but a quick illustration in terms of the classical Ising chain as a target Hamiltonian would be quite illuminating for readers who are amateurs in this area.

4-Four figures mention resampling using bootstrap. A brief explanation of this procedure before Sec.3 would add valuable clarity to the numerical results.

Report

The manuscript discusses a method of solving generic systems of constraints using variational states. This is done by building a target Hamiltonian whose ground state encodes for the solution with least number of broken constraints, and learning the desired state by optimizing the parameters of a variational wavefunction. Although the general technique is well established, the machinery often does not work well in practice for problems of interest due to a lack of powerful optimization protocols. The authors attempt to resolve this problem by adopting the technique of parallel tempering, which is borrowed from the field of Monte Carlo simulations of spin glasses. They expand the the system of interest to a set of Hamiltonians, which interpolate between the target Hamiltonian, and one which has an easy to train solution. This allows a high degree of exploration in the wavefunction space, thus leading to a higher chance of hitting the true ground state. They illustrate the performance of the algorithm using two representative problems, the first of which is chosen specifically to stymie the traditional version, and a second which is a toy problem from quantum chemistry.
The presentation is clear and concise, and all the included content is relevant for the point which the authors wish to make. Given that there is no constraint on page length, the manuscript would benefit from a more detailed description of the parallel tempering method, and from a short example which illustrates the changes in energy for each replica as a function of simulation time.
The manuscript meets all of the general acceptance criteria, and meets the expectation of "Open a new pathway in an existing or a new research direction, with clear potential for multipronged follow-up work". The method is of interest to the adiabatic quantum computing community as a whole, and one can expect that many researchers in the field will attempt to build on it.

Requested changes

1-Given that the entire manuscript deals with an RBM architecture, it is essential that this be mentioned in the abstract.

2-Typo: Above Eq.8, "linearly" twice

3-An intuition about the performance of the method can be provided by studying the behavior of the energy and variance during simulation, this will help the general reader understand the technique better.

4-The discussion below Eq.14 is missing the definition of $n$, and the description of selecting the appropriate $u$ is too terse. If possible, a short example (maybe the classical Ising chain) can be included.

5-The resampling procedure using bootstrap is not described in the main text, this should be included as a short note.

6-A brief discussion about the effect of the number of replicas and how much the ground states of various replicas differ, and how this affects the performance should be included. Perhaps this is best done in Sec. 3A using the language of the precipice problem.

7-After Eq.15, one should use $H_T(s)$ instead of $H(s)$.

8-In the conclusion section, the authors equate stoquasticity with ease of representation. This should be discussed in a bit more detail, as it is not immediately apparent why this is so from the RBM architecture, which is built in terms of complex parameters.

9-In Eq.A6, $S_{k,k^{\prime}}$ instead of $s_{k,k^{\prime}}$.

---

## Round 2 · Referee Report · Anonymous (Referee 2) · 2022-12-21

Strengths

1- Introduction of a new and potentially widely applicable approach to escape local minima in variational Monte Carlo 2- Systematic benchmarking of the new approach that demonstrates its merit 3- Very clear and structured presentation

Weaknesses

1- Study restricted to system sizes where exact methods are applicable 2- Role of the choice of the replica swap probability remains unclear 3- Source code not available

Report

The authors address the issue of getting stuck in local minima that arises in variational ground state searches with neural network ansatz functions. In order to remedy this problem, they introduce a new optimization method inspired by quantum parallel tempering: Multiple replicas of the ansatz are optimized with varying admixtures of a "mixer Hamiltonian" and the replicas are regularly swapped during the procedure. Using two model Hamiltonians, where it is known that finding ground states is difficult due to false minima, they demonstrate that the parallel tempering approach can have a clear advantage over conventional Stochastic Reconfiguration.

As neural network quantum states become more and more widely used, this work addresses a very timely topic and it introduces an original and promising new component to the toolbox. This tool can be used by follow-up works as presented in the manuscript, but, as the authors point out, also its better understanding and further optimization opens different directions for future work. Thereby, the manuscript meets the expectations for acceptance in SciPost Physics.

The paper also meets the general acceptance criteria 1-4 and 6. Regarding criterion 5 (reproducibility), the manuscript seems to contain all necessary information. But, ideally, the used source code should be made available as it is common practice in the machine learning community.

The presented study is very well organized, the data is convincing, and the presentation is clear. There are only two points of criticism: First, the work is restricted to small system sizes and tailored model systems, where exact solutions are available. It would be interesting to see whether the parallel tempering approach yields an advantage in a larger system of physical interest, e.g. the J1-J2 model that has become a common testing ground in the community. Second, the replica swap probability used by the authors is motivated by physical intuition, but the choice is still arbitrary. A comparison with some trivial choice for the swap probability, e.g. uniform, could give an idea how large the influence of the choice is on the result.

Finally, one suggestion for first additional insights into the efficiency of the parallel tempering would be to look at trajectories of individual replicas during the optimization. This would reveal how often replicas are swapped and whether they are passed all the way between the target and the mixer Hamiltonian.

In my opinion, the manuscript is in good shape to be published in SciPost Physics after a minor revision.

Requested changes

1- Formatting needs to be changed to SciPost template. 2- Create a public repository containing the source code. 3- Consider the two main points of criticism mentioned above for a revised version. 4- Although it appeared only two days earlier on the preprint server, I would suggest to add also https://arxiv.org/abs/2211.07749 in the introduction besides Refs [16,17], because it constitutes a significant step forward beyond Ref. [16].

---

## Round 3 · Referee Report · Pranay Patil · 2023-2-6

Report

In their updated manuscript, the authors have improved their presentation of the subject matter considerably, leading to significantly more clarity. Although a thorough scaling analysis of the efficiency of the proposed algorithm would be ideally desired, this appears to be outside the scope of the current work, and the authors have already included samples at different system sizes which clearly illustrates the qualitative advantage which the parallel tempering method provides over simple optimization. The authors have addressed all the referee suggestions satisfactorily.

---

## Round 3 · Referee Report · Anonymous · 2023-2-17

Strengths

See previous report.

Report

All questions raised in the previous report were answered. I only found that the second last paragraph of the Conclusions was not adjusted to the changes in Eq. (8) of the new version. Once this is fixed, I recommend publication in SciPost Physics.

Requested changes

1- Adjust the Conclusions according to the changed Eq. (8).

---

## Round 3 · Author Response

Dear Editor, Thank you for sharing with us the report of the Referees, which were generally very positive. While addressing the comments and requests the Referees made, we identified a simpler quantum parallel tempering scheme with even better performance than the method presented in our original submission. We have therefore relegated our original method to an appendix and updated the main text with the improved results. We address below point-by-point the comments and requests made by the Referees. We hope revised manuscript has satisfactorily addressed all the points made by the Referees. Thank you for your consideration.

Re: Report 1 by Pranay Patil We paste below verbatim the full transcript provided to us, interspersed by our responses and action taken on each point. We thank the Referee for taking the time to prepare their report. We believe that all concerns have been satisfactorily addressed in the response and/or manuscript.

  • Referee: The manuscript discusses a method of solving generic systems of constraints using variational states. This is done by building a target Hamiltonian whose ground state encodes for the solution with least number of broken constraints, and learning the de- sired state by optimizing the parameters of a variational wavefunction. Although the general technique is well established, the machinery often does not work well in practice for problems of interest due to a lack of powerful optimization protocols. The authors attempt to resolve this problem by adopting the technique of parallel tempering, which is borrowed from the field of Monte Carlo simulations of spin glasses. They expand the the system of interest to a set of Hamiltonians, which interpolate between the target Hamiltonian, and one which has an easy to train solution. This allows a high degree of exploration in the wavefunction space, thus leading to a higher chance of hitting the true ground state. They illustrate the performance of the algorithm using two representative problems, the first of which is chosen specifically to stymie the traditional version, and a second which is a toy problem from quantum chemistry. The presentation is clear and concise, and all the included content is relevant for the point which the authors wish to make. Authors: We thank the referee for their positive comments.

  • Referee: Given that there is no constraint on page length, the manuscript would benefit from a more detailed description of the parallel tempering method, and from a short example which illustrates the changes in energy for each replica as a function of simulation time. Authors: In order to illustrate how the parallel tempering works, we have included a new discus- sion and a new pair of figures (Figs. 3(a) and (b)) in Section 3.1 that gives an example of a random walk done by the replicas and how it can help the simulation escape from local minima.

  • Referee: The manuscript meets all of the general acceptance criteria, and meets the expectation of “Open a new pathway in an existing or a new research direction, with clear potential for multipronged follow-up work”. The method is of interest to the adiabatic quantum computing community as a whole, and one can expect that many researchers in the field will attempt to build on it. Authors: We thank the referee for their positive recommendation.

Requested changes 1. Given that the entire manuscript deals with an RBM architecture, it is essential that this be mentioned in the abstract. We now explicitly say that our examples use Restricted Boltzmann Machines as their parameterized ANN.

  1. Typo: Above Eq.8,“linearly” twice The text associated with this has been moved to an Appendix and has been fixed.

  2. An intuition about the performance of the method can be provided by studying the behavior of the energy and variance during simulation, this will help the general reader understand the technique better. We have included a new discussion and a new pair of figures (Figs. 3(a) and (b)) in Section 3.1 that gives an example of a random walk done by the replicas and how it can help the simulation escape from local minima.

  3. The discussion below Eq.14 is missing the definition of n, and the description of selecting the appropriate u is too terse. If possible, a short example (maybe the classical Ising chain) can be included. We have now included the definition of n as being the total number of qubits. Did the referee means s instead of u? If so, we have elaborated further on the problem. We hope the expanded description is more clear.

  4. The resampling procedure using bootstrap is not described in the main text, this should be included as a short note. We have included a new appendix (Appendix E) that describes the bootstrap procedure.

  5. A brief discussion about the effect of the number of replicas and how much the ground states of various replicas differ, and how this affects the performance should be included. Perhaps this is best done in Sec. 3A using the language of the precipice problem. We have elaborated further on the role of the number of replicas N on the performance for the Precipice problem at the end of section (now) 3.1. Specifically, we see that at intermediate system sizes, the addition of more replicas beyond N = 10 does not provide any additional advantage, but at the largest system sizes, we clearly need more replicas to find the ground state.

  6. After Eq.15, one should use HT (s) instead of H(s). Fixed.

  7. In the conclusion section, the authors equate stoquasticity with ease of representation. This should be discussed in a bit more detail, as it is not immediately apparent why this is so from the RBM architecture, which is built in terms of complex parameters. We have elaborated further on this comment. Our point was not necessarily about the ease of representation using RBM’s, but more that stoquastic Hamiltonians are often suggested to be easier to solve than their non-stoquastic counterparts.

  8. In Eq.A6, S_{k,k′} instead of s_{k,k′} . Fixed.

Re: Anonymous Report 2 We paste below verbatim the full transcript provided to us, interspersed by our responses and action taken on each point. We thank the Referee for taking the time to prepare their report. We believe that all concerns have been satisfactorily addressed in the response and/or manuscript.

  • Referee: The authors address the issue of getting stuck in local minima that arises in varia- tional ground state searches with neural network ansatz functions. In order to remedy this problem, they introduce a new optimization method inspired by quantum parallel tempering: Multiple replicas of the ansatz are optimized with varying admixtures of a “mixer Hamiltonian” and the replicas are regularly swapped during the procedure. Us- ing two model Hamiltonians, where it is known that finding ground states is difficult due to false minima, they demonstrate that the parallel tempering approach can have a clear advantage over conventional Stochastic Reconfiguration. As neural network quantum states become more and more widely used, this work ad- dresses a very timely topic and it introduces an original and promising new component to the toolbox. This tool can be used by follow-up works as presented in the manuscript, but, as the authors point out, also its better understanding and further optimization opens different directions for future work. Thereby, the manuscript meets the expectations for acceptance in SciPost Physics. Authors: We thank the referee for their positive review.

  • Referee: The paper also meets the general acceptance criteria 1-4 and 6. Regarding criterion 5 (reproducibility), the manuscript seems to contain all necessary information. But, ideally, the used source code should be made available as it is common practice in the machine learning community. Authors: We now include a link to our personal Github repository with the source code to reproduce the results in the main text.

• Referee: The presented study is very well organized, the data is convincing, and the presentation is clear. There are only two points of criticism: First, the work is restricted to small system sizes and tailored model systems, where exact solutions are available. It would be interesting to see whether the parallel tempering approach yields an advantage in a larger system of physical interest, e.g. the J1-J2 model that has become a common testing ground in the community. Authors: We understand the Referee’s point about this. Our goal is to eventually scale our simulations to system sizes for which exact methods fail, but our aim with this work was to first demonstrate that the methodology works before introducing additional complications. For example, at larger system sizes, exact sampling is no longer feasible, and finite sampling introduces other issues to the training that we wanted to disentangle from the study of the methodology. Furthermore, at larger system sizes, the choice of ANN architecture becomes more crucial, and again this was a detail we wanted to avoid at this early stage.

  • Referee: Second, the replica swap probability used by the authors is motivated by physical intuition, but the choice is still arbitrary. A comparison with some trivial choice for the swap probability, e.g. uniform, could give an idea how large the influence of the choice is on the result. Authors: We thank the referee for bringing up this point. While trying to address this point, we identified a simpler and more better performing swap probability, which is now the method presented in the main text. The new approach is similar in spirit to using the free energy for the local energy calculations, where instead of the entropy, we use the expectation value of the mixer Hamiltonian. We have relegated our original method to the appendix and also include an almost-fixed swap probability rule. All the methods presented outperform the non-QPT case, but they still have differences between them.

  • Referee: Finally, one suggestion for first additional insights into the efficiency of the parallel tempering would be to look at trajectories of individual replicas during the optimization. This would reveal how often replicas are swapped and whether they are passed all the way between the target and the mixer Hamiltonian. Authors: We have included a new discussion and a new pair of figures (Figs. 3(a) and (b)) in Section 3.1 that gives an example of a random walk done by the replicas and how it can help the simulation escape from local minima.

Requested changes 1. Formatting needs to be changed to SciPost template. Done.

  1. Create a public repository containing the source code. Done.

  2. Consider the two main points of criticism mentioned above for a revised version.. We have implemented one of them (an example of the individual trajectories during optimization) and hope we can address the second (the J1−J2 model at larger system sizes) in future work.

  3. Although it appeared only two days earlier on the preprint server, I would suggest to add also https://arxiv.org/abs/2211.07749 in the introduction besides Refs [16,17], because it constitutes a significant step forward beyond Ref. [16]. We have included this new reference.

---

## Round 3 · List of Changes

- We have updated the formatting of the manuscript to the SciPost template.
- We have updated the QPT scheme used in the main text (the new Eqts. (7) and (8)). The new scheme is simpler and better performing, as demonstrated by the updated Figs. (2), (4), (5), and (6). The previous QPT scheme is now presented in Appendix D.1, and we have included another QPT scheme in Appendix D.2.
- We have elaborated further on the QPT method and included a new figure (Fig.(3)) that illustrates how it works.
- We have included an appendix (Appendix E) describing the bootstrap procedure used.
- We have adjusted to abstract to explicit mentioned that we focus entirely on Restricted Boltzmann Machines as the architecture of our artificial neural network ansatz.
- We have elaborated on our comment about stoquasticity in the Conclusion section.
- We have corrected the typos pointed out by the Referees.
- We have included additional references.

---

## Round 4 · Author Response

Dear Editor, Thank you for sharing with us the report of the Referees. We have adjusted the second to last paragraph in the Conclusions to be consistent with the new version of the manuscript. We also identified a typo in Eqt. 10 that we have fixed. We believe that addresses the remaining issues raised by the referees.

---

## Round 4 · List of Changes

• Changed 'linear' to 'cubic' in the 2nd to last paragraph of the Conclusions.
  • Fixed a typo in Eqt. 10.

---

## Editorial Decision

published